# Carbon monoxide as a cellular protective agent in a swine model of cardiac arrest protocol

John C. Greenwood[1], Ryan W. Morgan[2,3], Benjamin S. Abella[1], Frances S. Shofer[1], Wesley B. Baker[2,4,5], Alistair Lewis[5,6], Tiffany S. Ko[2,3], Rodrigo M. Forti[4,5], Arjun G. Yodh[7], Shih-Han Kao[2], Samuel S. Shin[8], Todd J. Kilbaugh[2,3], David H. Jang[1,2]*

1 Department of Emergency Medicine, Perelman School of Medicine, University of Pennsylvania, Philadelphia, PA, United States of America, 2 Resuscitation Science Center, The Children's Hospital of Philadelphia, Philadelphia, PA, United States of America, 3 Department of Anesthesiology and Critical Care Medicine, Children's Hospital of Philadelphia, Philadelphia, PA, United States of America, 4 Department of Neurology, Perelman School of Medicine, University of Pennsylvania, Philadelphia, PA, United States of America, 5 Division of Neurology, Children's Hospital of Philadelphia, Philadelphia, PA, United States of America, 6 Department of Chemistry, University of Pennsylvania, Philadelphia, PA, United States of America, 7 Department of Physics and Astronomy, University of Pennsylvania, Philadelphia, PA, United States of America, 8 Department of Pharmacology, Perelman School of Medicine, University of Pennsylvania, Philadelphia, PA, United States of America

* david.jang@pennmedicine.uphs.edu

**Data Availability Statement:** No datasets were generated or analysed during the current study. All relevant data from this study will be made available upon study completion.

## Abstract

Out-of-hospital cardiac arrest (OHCA) affects over 360,000 adults in the United States each year with a 50–80% mortality prior to reaching medical care. Despite aggressive supportive care and targeted temperature management (TTM), half of adults do not live to hospital discharge and nearly one-third of survivors have significant neurologic injury. The current treatment approach following cardiac arrest resuscitation consists primarily of supportive care and possible TTM. While these current treatments are commonly used, mortality remains high, and survivors often develop lasting neurologic and cardiac sequela well after resuscitation. Hence, there is a critical need for further therapeutic development of adjunctive therapies. While select therapeutics have been experimentally investigated, one promising agent that has shown benefit is CO. While CO has traditionally been thought of as a cellular poison, there is both experimental and clinical evidence that demonstrate benefit and safety in ischemia with lower doses related to improved cardiac/neurologic outcomes. While CO is well known for its poisonous effects, CO is a generated physiologically in cells through the breakdown of heme oxygenase (HO) enzymes and has potent antioxidant and anti-inflammatory activities. While CO has been studied in myocardial infarction itself, the role of CO in cardiac arrest and post-arrest care as a therapeutic is less defined. Currently, the standard of care for post-arrest patients consists primarily of supportive care and TTM. Despite current standard of care, the neurological prognosis following cardiac arrest and return of spontaneous circulation (ROSC) remains poor with patients often left with severe disability due to brain injury primarily affecting the cortex and hippocampus. Thus, investigations of novel therapies to mitigate post-arrest injury are clearly warranted. The primary objective of this proposed study is to combine our expertise in swine models of CO and cardiac arrest for

**Funding:** Financial disclosure (grants): 1. National Institute of Environmental Health Sciences R21ES031243 (Jang) 2. National Heart, Lung, and Blood Institute R01HL141386 (Kilbaugh) 3. National Heart, Lung, and Blood Institute R56HL158696 (Jang) 4. National Heart, Lung, and Blood Institute R01HL166592 (Jang, Baker) 5. Children's Hospital of Philadelphia Frontier Program (Baker, Ko, Forti, Kilbaugh) 6. National Institute of Neurologial Disorders and Stroke R01NS113945 (Baker, Lewis, Forti, Kilbaugh) 7. U. S. Department of Defense DoD W81XWH-22-1-0887/8 (Baker, Ko, Forti, Kilbaugh) 8. Toyota Way Forward Fund (Baker, Ko, Forti, Kilbaugh) The funders did not and will not have a role in study design, data collection and analysis, decision to publish, or preparation of the manuscript.

**Competing interests:** The authors have declared that no competing interests exist.

future investigations on the cellular protective effects of low dose CO. We will combine our innovative multi-modal diagnostic platform to assess cerebral metabolism and changes in mitochondrial function in swine that undergo cardiac arrest with therapeutic application of CO.

## Introduction

Out-of-hospital cardiac arrest (OHCA) affects over 360,000 adults in the U.S. each year with a 50–80% pre-hospital mortality [1, 2]. Despite initial resuscitation and aggressive post-arrest care following return of spontaneous circulation (ROSC), half of adults do not live to hospital discharge and less than 7% of patients have good neurologic recovery. OHCA ranks third ($10.2B) in annual economic productivity loss in the U.S. behind cancer and chronic heart disease [3, 4]. This equates to an economic loss of about $3,750 per U.S. taxpayer family. The mechanisms of cardiac arrest injury are complex and involve multiple pathways that include organ hypoxia, inflammation, metabolic derangements, and mitochondrial dysfunction [5–7]. The brain and the heart have the highest energy demand and are dependent on healthy mitochondria for both cellular homeostasis and post-arrest recovery. Hence, impaired mitochondrial function leads to metabolic crisis in critical organs which increases the risk of poor post-arrest outcomes. While multiple studies have implicated the mitochondria as key mediators of post-arrest brain injury, the role is less defined and warrant further investigation [8, 9].

As mitochondrial dysfunction may have a central role in cardiac arrest and the following neurological injury that occurs in survivors, therapies that may have targeted effects in the mitochondria are needed. One such therapeutic agent with promise to improve mitochondrial function with clinical benefit is carbon monoxide (CO). CO is mostly known for its adverse effects and is considered a leading cause of death from environmental exposures [10–12]. The adverse effects of CO include cellular hypoxia by the formation of carboxyhemoglobin (COHb) and mitochondrial dysfunction through the inhibition of Complex IV (CIV) leading to decreased ATP and increased ROS [13–15]. Despite the adverse effects of substantial CO exposure, CO is also generated physiologically in cells through the breakdown of reactive heme molecules by the enzyme heme oxygenase (HO) that has potent antioxidant and anti-inflammatory activities [16–18]. HO catalyzes the breakdown of heme into iron, biliverdin and CO. It has been shown that in cardiac ischemia there is an up-regulation of HO and an important mediator of ischemic-reperfusion (IR) injury [19–21]. While the removal of cyto-toxic heme is thought to play a protective role, CO has also been shown to have a therapeutic effect and in low doses may prevent excessive ROS production [20]. For example, increasing endogenous CO has been shown to decrease infarct size in acute myocardial infarction (MI) with a reduction in cell death and increased mitogenesis [22].

While CO has been studied in myocardial infarction itself, the role of CO in cardiac arrest and post-arrest care as a therapeutic is less defined despite supporting experimental data that show benefit [23–25]. Currently, the standard of care for post-arrest patients consists primarily of supportive care and TTM [26, 27]. Despite current standard of care, the neurological prognosis following cardiac arrest and return of spontaneous circulation (ROSC) remains poor with patients often left with severe disability due to brain injury primarily affecting the cortex and hippocampus. Thus, investigations of novel therapies to mitigate post-arrest injury are clearly warranted.

The primary objective of this protocol is to leverage our combined expertise in swine models of CO and cardiac arrest for future investigations on the cellular protective effects of low

dose CO. We propose to combine our innovative multi-modal diagnostic platform to assess cerebral metabolism and mitochondrial function in swine that undergo cardiac arrest with therapeutic application of CO [28–30].

## Material and methods

### Large animal justification

A swine model was chosen because swine size, cardiovascular and neuroanatomy, physiological responses, and inflammatory responses result in outcomes most like humans with critical illness [31, 32]. The size of the pig was chosen as it closely mimics the development of an adult human in terms of both neurological (shape, gyral pattern, neurovasculature anatomy, and distribution of gray and white matter) and cardiac development [33–35]. There is also strong similarity between swine and a human's anterior-posterior chest diameter and chest compression characteristics, which are critical for cardiac arrest resuscitation experiments [36–39]. All these characteristics favor the use of the swine for cardiovascular assessment and pharmacological studies in cardiac arrest. Our lab has established expertise in swine models of CO poisoning and cardiac arrest with the relevant physiologic and biomolecular measures that will ensure successful execution of this study from a logistical and safety perspective [28, 29, 36, 40, 41].

### Animals and overall study design

This is a large animal protocol designed for future investigation of the therapeutic application of CO in our swine model of cardiac arrest using an experimental ventricular fibrillation (VF)-arrest approach. Yorkshire pigs (6 months, 30 kg) of equal sexes will be used for our proposed studies. All pigs that arrive will undergo an entrance exam that will include a baseline physical exam and assessment by veterinary staff and fecal occult testing for parasites. Animals will be acclimated for a minimum of one day prior to any experiments. All subject animals will be randomized to one of four groups prior to arrival. All animals will then be pre-medicated with 20 mg/kg ketamine, followed by inhaled isoflurane in 100% oxygen though a snout mask followed by endotracheal intubation and placement on a ventilator with additional procedures described below. All procedures are currently approved by the Institutional Animal Care and Use Committee at the Children's Hospital of Philadelphia (CHOP) and performed in accordance with the National Institutes of Health Guide for the Care and Use of Laboratory Animals for related studies with both CO and cardiac arrest in our lab.

### Perioperative procedures and monitoring

Following endotracheal intubation all animals will be placed on a mechanical ventilator. Ventilator settings will be as follows: tidal volume 10–11mL/kg, positive end-expiratory pressure 5 cm $H_2O$, and respiratory rate titrated to achieve an end-tidal of $CO_2$ 38–42 mmHg to minimize potential confounding changes in cerebral blood flow and acid–base status relevant for our non-invasive optical measurements of cerebral physiology. External jugular vein, femoral artery, and bilateral femoral veins will be cannulated with vascular introducer sheaths (Cordis Corp., Fremont, CA) under ultrasound guidance. The right femoral artery and vein access sites will be utilized for arterial pressure monitoring and central venous pressure monitoring, respectively. Isoflurane will then be weaned to approximately 0.5–1% to simulate human anesthetic protocols and minimize confounding toxicity and cerebral blood flow changes associated with higher doses of isoflurane while maintaining a surgical plane of anesthesia. A rectal temperature probe will be placed with normothermic temperature regulation from a warming

blanket. All data will be recorded with PowerLab 16/35 LabChart 8 Pro software from ADInstruments (Sydney, Australia). Arterial and venous blood samples will be drawn for serial lactate, $PCO_2$, cytokine, carboxyhemoglobin and serologic biomarker measurement. Continuous aortic pressure (MAP), central venous pressure (CVP), cardiac output (CO) and cardiac index (CI) will be monitored where CI is calculated by dividing CO by body surface area. A skin lead wire (St Jude Medical, Minnetonka, MN) will be advanced into the right ventricle and stimulate VF by direct current at 300 bpm to induce an R-on-T VF arrest [37, 38].

## Cardiac arrest and carbon monoxide experimental protocol

Our cardiac arrest protocol will consist of 8 min of untreated VF followed by standardized Advanced Cardiac Life Support (ACLS) consisting of cardiopulmonary resuscitation (CPR) with first defibrillation taking place 2 min after CPR is initiated (10 min after the start of the VF arrest) every two min until the ROSC or until 20 min of ACLS. Animals that achieve ROSC will be maintained under general anesthesia to a $PaO_2$ 60–100 mmHg, $PaCO_2$ 35–45 mmHg, and predefined hemodynamic targets with IV fluids to achieve adequate intravascular volume status, norepinephrine to achieve target mean arterial pressure (MAP), and epinephrine for a target CI. Normothermia and continuous hemodynamic monitoring will be maintained throughout the post-resuscitation experimental period. After successful resuscitation, the animals will be randomly divided into four groups: (1) CPR: Animals in this group will only receive CPR without CO upon ROSC; (2) CPR and CO: Cardiac arrest followed by CO treatment; (3) Sham: The same operation but no cardiac arrest or CO being given; (4) CO alone: Will be the same as Control except CO will be administrated to assess possible adverse effects of CO alone [Fig 1].

The following flow diagram will serve as a general framework for this proposed study with corresponding times: The perioperative period will include the induction of anesthesia to allow for the described procedures such as placement of central lines, cMD catheter, etc that typically takes about 2 hr. The induction of cardiac arrest will take about 4 seconds with cardiac arrest being allowed to continue for 8 min before the initiation of CPR with the first defibrillation at 10 min. If no ROSC is achieved, resuscitation will continue for an additional 10 min for a total of 20 min of CPR. Once ROSC is achieved, CO treatment with 100 ppm will be administrated for 2 hr for a total of 3 hr post-ROSC.

The assigned CO dose will be administered with a CO tank (244 cf) at 0–10L/min using a regulator with flow meter from Airgas (Radnor Township, PA, USA) for 100 ppm. Medical air will be administered for controls. The CO concentration entering the endotracheal tube will be monitored using an Inspector CO detector with a 0–2000 ppm range (Sensorcon, New York, USA). Animals in the CO group will receive CO at their assigned dose of 100 ppm. Sedation will be maintained with the use of fentanyl (5 μg/kg/h) and dexmedetomidine (2 μg/kg/h) during the CO gas exposure with discontinuation of isoflurane once the exposure is initiated. Our previous prior work has utilized CO doses of 400 ppm and 2000 ppm.

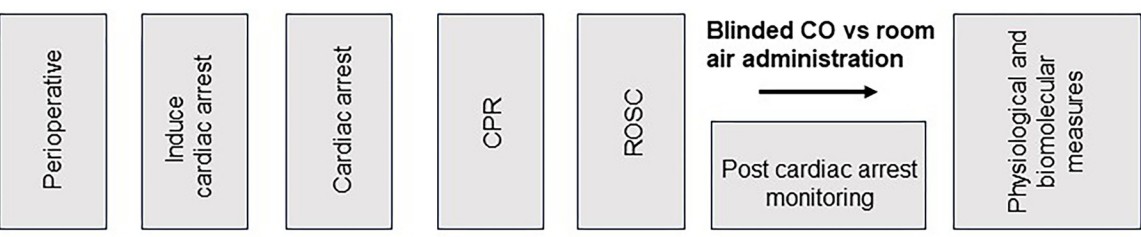

**Fig 1. Experimental protocol and flow diagram of study.**

## Cortical microcirculation imaging and cerebral blood flow measurement

After induction of anesthesia, the scalp will be locally infiltrated with 2% lidocaine and a left craniotomy (approximately 2x3 cm) will be created. Cortical microcirculation imaging will be performed using incident dark field (IDF) microscopy (CytoCam, Braedius Medical BV, Huizen, the Netherlands) with an intact dura to prevent any potential tissue necrosis. Video sequences will be obtained by placing the CytoCam device on the left frontotemporal area of the brain and maintained in position with a fixed support arm. Image acquisition and analysis will follow published consensus standards. Baseline images will be obtained after intubation and invasive hemodynamic monitoring is in place prior to cardiac arrest. Repeated measurements will be obtained at defined time points after ROSC. To ensure only vessels contributing to tissue gas exchange are included, only vessels < 20 μm in diameter will be included in IDF analysis [42–45].

## Non-invasive optical monitoring of cerebral oxygenation and CIV redox rates

Our research team has built a hybrid diffuse optical device that combines the techniques of frequency-domain diffuse optical spectroscopy (FD-DOS), diffuse correlation spectroscopy (DCS), and broadband diffuse optical spectroscopy (bDOS) to non-invasively monitor changes in cerebral oxygenation, blood flow, oxygen metabolism, and mitochondrial CIV (i.e., cytochrome c oxidase) redox state. Given that each tissue chromophore has a unique wavelength-dependent absorption signature, it is feasible to isolate the CIV influence on the brain's absorption spectrum from other tissue chromophores. We specifically used the UCLn algorithm to compute changes in cerebral oxidized CIV concentration from the measured wavelength-dependent changes in optical attenuation signals between 780–1000 nm wavelengths. To avoid cross-contamination between FD-DOS/DCS and bDOS, measurements will be temporally interleaved. Specifically, FD-DOS/DCS and bDOS data will be sequentially acquired for 1 minute each such that one complete set of measurements is obtained every 2 minutes. During bDOS acquisitions, tissue diffuse reflectance measurements are further interleaved with dark count measurements, *i.e.*, the broadband lamp shutter is programmatically opened and closed to take tissue and dark count spectral measurements, respectively (100 ms integration time for each spectrum). We have performed these methods in our previous studies and can be referenced for further details [28, 29].

## Measurement of cerebral microdialysis

Cerebral microdialysis (cMD) allows bedside semicontinuous monitoring of brain extracellular fluid for cerebral metabolism. cMD will be placed in the parietal cortex using a CMA 70 Elite from mDialysis (Stockholm, Sweden). Probes will be placed 10 mm deep in the brain parenchyma. Sterile saline will be perfused at 1 μl/min, and after a 30 min equilibration period, samples will be collected in 30 min intervals and samples will be immediately frozen at -80˚C. Pyruvate, lactate, glycerol and glucose concentrations will be analyzed in a blind fashion using the automated ISCUS FlexTM Microdialysis Analyzer and data will process using the ICUpilot software from mDialysis. Lactate and pyruvate values will be used to calculate a lactate-to-pyruvate ratio (LPR) to assess for redox balance. We have extensive experience using this cMD in various animal models of critical illness related to both cardiac arrest and CO poisoning [28, 29, 40, 46, 47].

## Tissue extraction and preparation

**Isolated brain mitochondria.** Upon completion of the protocol described above, animals will be euthanized with potassium chloride prior to brain tissue collection. Brain tissue will

then immediately undergo rapid but gentle dissection into 5 mm coronal slices. Part of the brain tissue will be snap frozen for later western blotting and ATP analysis. Additionally, left frontal cortical and left hippocampal tissue samples will be collected (to obtain isolated mito-chondria; described below) and placed into ice-cold isolation buffer solution (320 mM sucrose, 2 mM EGTA, 10 mM Trizma base, pH 7.4). Brain tissue will also be transferred into ice-cold 1X brain buffer (made from 0.5 L of 2X buffer: 225 mM D-Mannitol, 75 mM Sucrose, 5 mM HEPES, 1 mM EGTA and 0.5 L of double deionized water, pH 7.4), manually homogenized in 0.2% BSA buffer (catalog A6003) and be centrifuged at 1,300g and 4°C to separate the fatty pel-let from supernatant. The supernatant will then be centrifuged for 10 min at 21,000g to extract pellet. Brain mitochondria will be isolated from the derived pellet by differential centrifugation and application of density gradients using 15%, 23% and 40% Percoll (GE Healthcare cat. no. 17089101). Protein count for isolated mitochondria will finally be obtained with a Pierce BCA Protein Assay kit (catalog 23227) from Thermo Fisher Scientific (Waltham, MA, USA).

## Measurement of mitochondrial respiration in isolated brain mitochondria

Mitochondrial respiratory function will be analyzed using Oroboros O2k-FluoRespirometers (Oroboros Instruments, Innsbruck, Austria) with a substrate–uncoupler–inhibitor titration (SUIT) protocol. The SUIT protocol measures oxidative phosphorylation capacity with elec-tron flow through all components of the electron transport system. All data will be acquired using DatLab 7 (Oroboros Instruments, Innsbruck, Austria) and respiration value will be nor-malized to protein count for the isolated mitochondria of both cortical and hippocampal tissue with more details in our previous publications [28, 29, 40].

## Measurement of reactive oxygen species in isolated brain mitochondria

Measurements of ROS generation as hydrogen peroxide (converted superoxide) will be mea-sured using the O2k-Fluorescence LED2-Module attached to the Oroboros O2k-FluoRespi-rometer, permitting simultaneous measurements of hydrogen peroxide ($H_2O_2$) production and mitochondrial respiration, utilizing an Amplex UltraRed assay. In short, Amplex UltraRed (N-acetyl-3, 7 dihydroxyphenoxazine) (5 mM), in the presence of horseradish peroxidase (1 U/ml), reacts with $H_2O_2$ to produce the fluorescent compound resorufin. The addition of superoxide dismutase (SOD) (10 U/ml) ensures that all superoxide is converted into $H_2O_2$. A 3-point calibration of the fluorometric signal will be done prior to each measurement by the addition of 100 nM $H_2O_2$. Mitochondrial ROS generation is the predominant source of ROS and leads to alterations in redox signaling, oxidative damage to proteins and lipids, additional mitochondrial dysfunction and ultimately a major cause of ongoing secondary brain and heart injury [40].

## Western blot

Western blot will be performed on tissue with all reagents and antibodies purchased from Invi-trogen (Carlsbad, CA, USA) unless otherwise noted and will follow previous methods from our prior publication with some modification for specific antibodies [29].

The following will be performed to obtain protein quantification of Complex IV (using the subunit IV). Gel proteins will be transferred onto a PVDF membrane (catalog IB24001) and then incubated with a complex IV monoclonal antibody (catalog #A21348) with a dilution fac-tor of 1:4000 in iBind solution (Catalog SLF1020). Primary mouse monoclonal anti-GAPDH antibody (GA1R) conjugated to HRP 1:1333 (MA5-15738) will be used as an internal control. Complex IV protein concentrations will be detected using rabbit anti-mouse IgG secondary antibody conjugated to HRP (catalog A16160, 1:1600) and a chemiluminescent substrate

reagent kit. Immunoblotting steps will be done in an iBind Western Device (Invitrogen). iBright Analysis Software (Thermo Scientific) will be used in the quantification and densitometric analysis of the blots. Based on previous work we will use a protein concentration of 10 μl per well. All experiments will be performed in duplicates and the local background corrected density values will be normalized against GAPDH (CS) and GA1R (Complex IV) values.

## ATP fluorometry

ATP concentrations will be obtained in snap frozen brain tissue samples (ratio of 10 mg of tissue to 100 μl of the assay buffer) using an ATP fluorometric assay kit (Sigma MAK190) with an excitation of 535 nm and emission of 587 nm obtained in a similar manner based on our previous publications [40, 47].

## Measurement of inflammation

Plasma will be isolated from whole blood samples every half hour by centrifugation and evaluated by multiplex enzyme immunoassay using the Q-Plex Porcine Cytokine Panel (4-Plex) using a multiplex ELISA Quansys Biosciences (Logan, Utah, USA) for the cytokines interleukin 1ß (IL-1ß; Pro-Tumor Inflammation), interleukin-6 (IL-6; a pro-inflammatory cytokine), interleukin-8 (IL-8; an inflammatory cytokine), and tumor necrosis factor (TNFα; inflammatory cytokine and acute phase reactant) [48].

## Statistics and data analysis

Continuous variables characterizing demographical data, microcirculation data, and mitochondrial respiration measurements, and outcomes data will be reported as means with standard deviations if normally distributed or medians with interquartile ranges if not normally distributed. Categorical variables will be represented as frequencies and proportions. To examine the predictive performance of selected variables for the primary outcome, we will construct receiver operator characteristic curves for threshold values of PVD, MHI, lactate, SvO2, MAP, and CI. A Youden index will be calculated to determine the best cutoff value for determining prolonged VVFDs. Linear regression modeling will be used to examine the relationship between L/P ratio and postoperative microcirculation variables. We will perform univariate analyses on candidate predictor variables of L/P ratio including PVD, MHI, LFTs, creatinine, CPB time, cross clamp time, and catecholamine administration. Multiple linear regression analysis will be used to model the effect of significant predictors. Repeated measure ANOVA will be used to compare changes in microcirculatory variables, mitochondrial respiration, and mitochondrial reactive oxygen species production over time. To adjust for multiple comparisons, post-hoc pairwise Tukey Kramer t-tests will be performed. All analyses will use statistical software (SAS version 15.1, Cary, NC; Prism v 9.0, Graph-Pad Software, San Diego, CA).

## Discussion

CO has traditionally been thought of as a cellular poison causing adverse effects from the combination of hypoxia, increased inflammation, and mitochondrial dysfunction [13–15]. In tissues that are highly sensitive to hypoxia such as the heart and the brain, affected patients can develop cardiac and neurologic symptoms that can lead to long-term morbidity in severe cases [11, 49]. While patients typically manifest symptoms with higher concentrations of CO, our lab have demonstrated cellular dysfunction in our swine model of acute CO poisoning using a relative low dose (400 ppm) [40]. Despite the known adverse effects of CO, low dose CO has

been shown to have therapeutic benefit in much lower doses in clinically relevant diseases such as acute MI and pulmonary disease [23, 24, 50–52].

While CO poisoning is the result of exogenous sources such as combustion and industry, CO is also endogenously produced through the metabolism of heme by HO to biliverdin, iron and CO. HO is primarily responsible in maintaining damage control and promoting cellular repair. The absence of HO through knockouts have been shown to increase susceptibility to ischemia from the combination of excessive heme that is highly oxidative and also from the lack of CO [18, 20, 53]. Studies have demonstrated the more precise role of CO as well as the protective effect of CO in IR injury. One preclinical study demonstrated that the exogenous administration of CO in myocardial infarction leads to reduced infarct size and reduced apoptosis in the absence of HO. Other studies have also shown \ increased mitochondrial biogenesis from CO signaling as an important mechanism.

Another potential application for CO is as a therapeutic following ROSC to mitigate IR injury. While CO has been used therapeutically with MI and pulmonary disease, there is limited clinical data with its use in cardiac arrest [25, 54]. Experimental models of cardiac arrest have used CO as a cellular protective agent in cardiac arrest [25, 54]. In limited animal models, the use of low dose CO following ROSC have demonstrated increased survival, improved mitochondrial function with both increased mitogenesis (production of new mitochondria) and mitochondrial autophagy (removal of damaged mitochondrial) as a housekeeping mechanism [23, 24]. While there are limited small animal models in this area, there is a paucity of large animal models that recapitulate what can occur in patients [25, 54]. Our prior work with swine demonstrates a highly translational model that better captures the physiology and biomolecular findings in patients. Leveraging our swine models of CO exposure and cardiac arrest, positive findings from this study would allow more rapid clinical translation in the future [28, 29, 40]. Taken together, current experimental data demonstrate that CO treatment may improve both survival and neurological outcome following cardiac arrest.

One of the primary concerns for the therapeutic application of CO is safety for both the patient and healthcare staff. Since CO is a gas, earlier clinical studies demonstrated that controlled low doses of inhaled CO can be administered safely and achieve predictable COHb concentrations. There were two clinical studies in which low dose CO (range of 100–200 ppm) were administered in short daily exposures (range of 70 to 120 min) over a period ranging from four to five consecutive days [50, 55, 56]. While these studies investigated the therapeutic effect for pulmonary disease, there were no significant differences in adverse events between the control group and the CO exposure group. Another study used a one-time dose of a considerable higher CO concentration of 500 ppm for 1 hr resulting in a COHb of 7% with the only complaint being a headache and no other adverse effects were noted [52].

Other agents with similar properties to CO have also been investigated, including nitric oxide (NO) and hydrogen sulfide ($H_2S$). $H_2S$ is also a gas with overlapping properties to CO found in the gas industry and sewage handling [57]. While $H_2S$ have been shown to have cardioprotective benefits, it is considerably more potent than CO with a very low therapeutic index that makes clinical implementation challenging [58–62]. NO is produced endogenously with important biological functions that involve vasodilation and angiogenesis [57, 63]. NO has clinically been used for the treatment of ischemic heart disease but also has been used post-arrest to reduce IR injury. Despite the potential benefits of NO, NO is reactive at higher doses that can cause increased ROS production leading to DNA damage and can also cause methemoglobinemia (MetHb). MetHb can worsen ischemia as the oxidized iron in the affected hemoglobin can no longer accept or deliver oxygen to tissue [57]. Taken together, while $H_2S$, NO and CO are all potentially poisonous, CO is not considered as reactive as NO and has considerably more robust experimental and clinical data supporting use of CO over $H_2S$ and NO with a more favorable safety profile.

There are varying formulations of CO that have been studied with advantages and disadvantages related to delivery and safety [64]. The most linear method of CO administration is through inhalation that has been safely utilized in several clinical trials described above. The advantage of using inhaled CO is that CO rapidly diffuses across the alveolar-capillary membrane and the same studies have demonstrated predictable and reliable induction of COHb generation. While there is concern with the use of a gas, our prior swine work using CO have safely utilized inhaled CO with the advantage of continuous monitoring CO concentration and the use of a bedside co-oximetry to obtain continuous COHb measurements similar to a standard pulse oximeter used for oxy/de-oxyhemoglobin monitoring [28, 29, 40]. Due to potential concerns with inhaled CO, other methods of CO delivery have been developed such as CO releasing molecules (CORMs). CORMs were developed to improve the safety and delivery of CO [65–67]. Using transition metals such as ruthenium to deliver CO,CORMs have also been designed to release CO under certain conditions so may provide targeted delivery although this is still being actively investigated. Despite these promising properties, a concern for clinical application is the toxicity of the metallics. and While preclinical evidence shows no major adverse effects, this is an important consideration if CORMs are to be implemented clinically [68, 69]. Other delivery agents include prodrugs and enteral delivery that also have experimental data in support for potential safer application [67, 70, 71]. Overall given the combination of efficacy and clinical safety with inhalation CO, this proposal will leverage our experience with this delivery method.

In summary, there is experimental data that demonstrate efficacy of CO as a therapeutic in animal models of cardiac disease with promising clinical safety data., To date there are no clinical trials that investigate the effects of exogenous CO in cardiac arrest. Our proposed research will utilize our robust swine model to investigate the therapeutic effects of CO in our cardiac arrest swine model using clinically relevant physiological, imaging, and biomolecular metrics.

## Author Contributions

**Conceptualization:** John C. Greenwood, Benjamin S. Abella.

**Data curation:** Frances S. Shofer, Alistair Lewis, Rodrigo M. Forti.

**Formal analysis:** Frances S. Shofer, Rodrigo M. Forti, Samuel S. Shin, Todd J. Kilbaugh.

**Funding acquisition:** Todd J. Kilbaugh, David H. Jang.

**Investigation:** Ryan W. Morgan, Benjamin S. Abella, Tiffany S. Ko, David H. Jang.

**Methodology:** Ryan W. Morgan, Wesley B. Baker, Alistair Lewis, Tiffany S. Ko, Arjun G. Yodh, Shih-Han Kao, Samuel S. Shin, Todd J. Kilbaugh, David H. Jang.

**Project administration:** John C. Greenwood.

**Software:** Arjun G. Yodh.

**Supervision:** Todd J. Kilbaugh, David H. Jang.

**Writing – original draft:** John C. Greenwood, Ryan W. Morgan, Benjamin S. Abella, Frances S. Shofer, Wesley B. Baker, Alistair Lewis, Tiffany S. Ko, Rodrigo M. Forti, Arjun G. Yodh, Shih-Han Kao, Samuel S. Shin, Todd J. Kilbaugh, David H. Jang.

**Writing – review & editing:** John C. Greenwood, Ryan W. Morgan, Benjamin S. Abella, Frances S. Shofer, Wesley B. Baker, Alistair Lewis, Tiffany S. Ko, Rodrigo M. Forti, Arjun G. Yodh, Shih-Han Kao, Samuel S. Shin, Todd J. Kilbaugh, David H. Jang.

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
