## [Decision Letter · Decision Letter 0]

2 Nov 2023

PONE-D-23-24722The feasibility of carbon monoxide as a cellular protective agent in a swine model of cardiac arrestPLOS ONE

Dear Dr. Jang,

Thank you for submitting your manuscript to PLOS ONE. After careful consideration, we feel that it has merit but does not fully meet PLOS ONE’s publication criteria as it currently stands. Therefore, we invite you to submit a revised version of the manuscript that addresses the points raised during the review process.

I would like to sincerely apologise for the delay you have incurred with your submission. It has been exceptionally difficult to secure reviewers to evaluate your study. We have now received two completed reviews; the comments are available below. Reviewer#1 has raised significant scientific concerns about the study that need to be addressed in a revision. About Reviewer#2 comments, I'd like to note here that Study Protocols are in scope for PLOS ONE (https://journals.plos.org/plosone/s/what-we-publish#loc-study-protocols).

Please revise the manuscript to address all the reviewer's comments in a point-by-point response in order to ensure it is meeting the journal's publication criteria. Please note that the revised manuscript will need to undergo further review, we thus cannot at this point anticipate the outcome of the evaluation process.

We look forward to receiving your revised manuscript.

Kind regards,

Miquel Vall-llosera Camps

Senior Staff Editor

PLOS ONE

2. Please amend your current title to include the word "protocol".

6. Please amend either the title on the online submission form (via Edit Submission) or the title in the manuscript so that they are identical.

8. Please upload a copy of Figure 1, to which you refer in your text. If the figure is no longer to be included as part of the submission please remove all reference to it within the text.

Reviewers' comments:

Reviewer's Responses to Questions

**Comments to the Author**

1. Does the manuscript provide a valid rationale for the proposed study, with clearly identified and justified research questions?

Reviewer #1: Yes

Reviewer #2: Yes

2. Is the protocol technically sound and planned in a manner that will lead to a meaningful outcome and allow testing the stated hypotheses?

Reviewer #1: Partly

Reviewer #2: Yes

3. Is the methodology feasible and described in sufficient detail to allow the work to be replicable?

Reviewer #1: Yes

Reviewer #2: Yes

4. Have the authors described where all data underlying the findings will be made available when the study is complete?

Reviewer #1: Yes

Reviewer #2: Yes

5. Is the manuscript presented in an intelligible fashion and written in standard English?

Reviewer #1: Yes

Reviewer #2: Yes

6. Review Comments to the Author

You may also provide optional suggestions and comments to authors that they might find helpful in planning their study.

Reviewer #1: Thank you for giving me an opportunity to review this manuscript. This manuscript describes a study protocol to investigate the feasibility of CO as a cellular protective agent in a pig model of cardiac arrest. Overall, the manuscript clearly presents the rationale and method of the study. I have only a few minor suggestions listed as follows.

Abstract

Abstract, Line 9. Please revise the term ‘post-cardiac patients’. (patients with cardiac arrest or patients resuscitated from cardiac arrest?)

Introduction

Introduction, Line 7. It may be better to move the sentence “In one of our studies, we found that swine exposed to 2000 ppm for over an hour reliable COHb of around 50%.3” from the introduction section to the method or discussion section.

Introduction. Please spell out all abbreviations including MI and CI in the first appearance. Please check all abbreviations throughout the manuscript.

Introduction. A more detailed description of why CO is a more ideal agent than NO or H2S is needed.

It is necessary to highlight the necessity of this study in relation to previous studies that investigated the effect of CO administration after arrest.

Materials and methods

The double quotation marks in the first sentence in the animals and overall study design section are incomplete.

Please provide a more detailed description on the CI monitoring.

The “up to 12 min” is unnecessarily repeated in the first sentence of the cardiac arrest and carbon monoxide experimental protocol section.

Please provide the rationale for choosing 8 min duration of untreated cardiac arrest.

Please provide the reason why epinephrine is chosen as a medication to titrate CI. Epinephrine is well known to affect cerebral perfusion adversely.

There are two sentences describing the timing of randomization: “All subject animals will be randomized to one of three groups prior to arrival.” and “After successful resuscitation, the animals will be randomly divided into three groups:”.

It would be helpful if you clearly described the timing of commencement of CO administration.

It would be nice if the authors could share their previous experiences of CO administration, such as hemodynamic changes after CO administration.

How many hours after ROSC does the experimental protocol end and euthanasia occur?

Please define the primary outcome of this study.

Figure legend

Please spell out all abbreviations within the Sources of Funding section.

Please revise the figure to include the timing of each intervention.

Reviewer #2: The proposed study is highly relevant with a very well described method. It will allow a very comprehensive evaluation of the effect of CO after cardiac arrest.

The submission of the design of experimental study should be encouraged. However, I do not know whether method paper could comply to Plos One Editorial policy, since the journal guidelines state that research articles "must present the results of original research". In order to include original findings, the article should at least present preliminary original data with a natural history of the model in control conditions.

7. PLOS authors have the option to publish the peer review history of their article (what does this mean?). If published, this will include your full peer review and any attached files.

Reviewer #1: No

Reviewer #2: No

---

## [Author Response · Author response to Decision Letter 0]

18 Dec 2023

Response to Reviewers

Reviewer #1: Thank you for giving me an opportunity to review this manuscript. This manuscript describes a study protocol to investigate the feasibility of CO as a cellular protective agent in a pig model of cardiac arrest. Overall, the manuscript clearly presents the rationale and method of the study. I have only a few minor suggestions listed as follows.

Abstract

1. Abstract, Line 9. Please revise the term ‘post-cardiac patients’. (patients with cardiac arrest or patients resuscitated from cardiac arrest?)

Response: We reviewed the manuscript and updated the term for post cardiac arrest patients to “post-arrest patients” which is the proper term. We revised both the abstract and the main manuscript. 

2. Introduction, Line 7. It may be better to move the sentence “In one of our studies, we found that swine exposed to 2000 ppm for over an hour reliable COHb of around 50%.3” from the introduction section to the method or discussion section.

Response: We move this sentence from the introduction to the methods as recommended by reviewer 1. We also restructured the introduction to focus more on cardiac arrest as well as the gaps related to the need for more therapies as opposed to starting with carbon monoxide poisoning that we feel flows better. 

3. Introduction. Please spell out all abbreviations including MI and CI in the first appearance. Please check all abbreviations throughout the manuscript.

Response: We spell out all abbreviations such as MI (myocardial infarction) and CI (cardiac index) in the manuscript in tracked changes. 

4. Introduction. A more detailed description of why CO is a more ideal agent than NO or H2S is needed.

Response: We have already described the differences between CO, NO and H2S in the discussion section that flows better here as opposed to placing in the introduction.

Discussion Paragraph 5

“Other agents with similar properties to CO have been investigated, including nitric oxide (NO) and hydrogen sulfide (H2S). H2S is also a toxic gas with overlapping properties to CO that also can be found in the gas industry and sewage handling. While H2S has been shown to have cardioprotective benefits, it is considerably more toxic than CO with a very low therapeutic index that makes clinical implementation challenging. Another gas that also has been used is NO that is produced endogenously with important biological functions that primarily involve vasodilation and angiogenesis. NO has clinically been used for the treatment of ischemic heart disease but also has been used post-arrest to reduce IR injury. Despite the potential benefits of NO, NO is reactive at higher doses that can cause increased ROS production leading to DNA damage and can also cause methemoglobinemia (MetHb). MetHb can worsen ischemia as the oxidized iron in the affected hemoglobin can no longer accept or deliver oxygen to tissue. Taken together, while H2S, NO and CO are all potentially poisonous, CO is not considered as reactive as NO and has considerably more robust experimental and clinical data supporting use of CO over H2S and NO with a more favorable safety profile.”

5. It is necessary to highlight the necessity of this study in relation to previous studies that investigated the effect of CO administration after arrest.

Response: We revised to better discuss the necessity of this study as there is a paucity of work in this area despite both in vitro and small animal work that support the of CO as a therapeutic in cardiac arrest. 

Discussion Paragraph 3

“While there are some small animal models in this area, there is a paucity of large animal models that recapitulate what can occur in patients. Our prior work with swine demonstrates a much more translational model that better captures the physiology and biomolecular findings in patients. Leveraging our swine models of both CO exposure and cardiac arrest, positive findings from this study would allow more rapid clinical translation in the future.”

Materials and methods

6. The double quotation marks in the first sentence in the animals and overall study design section are incomplete.

Response: This has been fixed as recommended.

7. Please provide a more detailed description on the CI monitoring.

Response: We revised the methods to provide a more detailed description of cardiac index or CI monitoring.

Perioperative procedures and monitoring section

“Continuous aortic pressure (MAP), central venous pressure (CVP), cardiac output (CO) and cardiac index (CI) will be monitored with CI is calculated by dividing CO by body surface area.”

8. The “up to 12 min” is unnecessarily repeated in the first sentence of the cardiac arrest and carbon monoxide experimental protocol section.

Response: This was corrected

9. Please provide the rationale for choosing 8 min duration of untreated cardiac arrest.

Response: Extensive preclinical and clinical research has examined the time-sensitive progression of resuscitation physiology. We first considered the 3 phase model of cardiac arrest which was previously described by Weisfeldt & Becker (JAMA 2002), where immediate defibrillation within a 5-minute arrest period did not benefit from CPR (Neimann et al, Circulation, 1992). For our proposed experiment, we wanted to mimic a real world scenario of sudden cardiac arrest followed by bystander CPR, while also maximizing our chances for return of spontaneous circulation. A number of clinical and preclinical studies have identified that ROSC rates can be optimized in the 5-10 minute no-flow window by including CPR + Defibrillation (Yakaitis et al., Crit Care, 1980; Menegazzi et al, Ann of Emerg Med, 1993; Yu et al, Circulation, 2002). Extending beyond the 10 minute no-flow time appears to decrease chances of ROSC, and the severity of brain injury (outcome of interest) will likely outweigh any benefit from the proposed intervention.

10. Please provide the reason why epinephrine is chosen as a medication to titrate CI. Epinephrine is well known to affect cerebral perfusion adversely.

Response: The post-cardiac arrest period is well known to result in reperfusion vasoplegia that would make it very difficult to initiate inodilator therapies (dobutamine, milrinone, etc.) without causing hemodynamic collapse. It is true that during ACLS, preclinical studies have identified that bolus dose (30 mcg/kg) epinephrine can reduce cerebral perfusion (Ristago, Crit Care Med, 2007), but the data is mixed, but our lab has found that lower dose (20 mcg/kg) may actually increase cerebral perfusion (Mavroudis et al, Crit Care, 2020). Post-arrest titration of an epinephrine effusion will allow us to augment cardiac output, while also avoiding excessive vasoplegia. It is unclear if epinephrine infusions, at lower continuous doses, negatively impact cerebral perfusion.

11. There are two sentences describing the timing of randomization: “All subject animals will be randomized to one of three groups prior to arrival.” and “After successful resuscitation, the animals will be randomly divided into three groups:”

Response: We revised to the following below

Material and methods under cardiac arrest and carbon monoxide experimental protocol

“After successful resuscitation, the animals will be randomly divided into four groups: (1) CPR: Animals in this group will only receive CPR without CO upon ROSC; (2) CPR and CO: Cardiac arrest followed by CO treatment; (3) Sham: The same operation but no cardiac arrest or CO being given; (4) CO alone: Will be the same as Control except CO will be administrated to assess possible adverse effects of CO alone [Figure 1].”

12. It would be helpful if you clearly described the timing of commencement of CO administration.

Response: We revised both the methods and Figure 1 to better describe the timing of each intervention so it is clear when CPR is induced followed by when CPR and CO administration occurs. For CO administration, an hour after achieving ROSC, CO (100 ppm) will be administered for 1 hour followed by 1 hour of re-oxygenation for a total post ROSC time of 3 hours. 

Figure 1: Experimental Protocol and flow diagram of study

The following flow diagram will serve as a general framework for this proposed study with corresponding times: The perioperative period will include the induction of anesthesia to allow for the described procedures such as placement of central lines, cMD catheter, etc that typically takes about 2 hr. The induction of cardiac arrest will take about 4 seconds with cardiac arrest being allowed to continue for 8 min before the initiation of CPR with the first defibrillation at 10 min. If no ROSC is achieved, resuscitation will continue for an additional 10 min for a total of 20 min of CPR. Once ROSC is achieved, CO treatment with 100 ppm will be administrated for 1 hr followed by 1 hr of re-oxygenation for a total of 3 hr post-ROSC.

13. It would be nice if the authors could share their previous experiences of CO administration, such as hemodynamic changes after CO administration.

Response: We revised the paper to cite our work both in the area of CO poisoning and cardiac arrest that demonstrates our expertise in this area.

“While patients typically manifest symptoms with higher concentrations of CO, our lab has demonstrated cellular dysfunction in our swine model of acute CO poisoning using a relative lower dose (400 ppm) without overt clinical symptoms”

14. How many hours after ROSC does the experimental protocol end and euthanasia occur?

Please define the primary outcome of this study.

Response: We revised the methods and Figure 1 to better describe the timing of each key event in our procedure.

Material and Methods

Figure 1: Experimental Protocol and flow diagram of study

“The following flow diagram will serve as a general framework for this proposed study with corresponding times: The perioperative period will include the induction of anesthesia to allow for the described procedures such as placement of central lines, cMD catheter, etc that typically takes about 2 hr. The induction of cardiac arrest will take about 4 seconds with cardiac arrest being allowed to continue for 8 min before the initiation of CPR with the first defibrillation at 10 min. If no ROSC is achieved, resuscitation will continue for an additional 10 min for a total of 20 min of CPR. Once ROSC is achieved, CO treatment with 100 ppm will be administrated for 1 hr followed by 1 hr of re-oxygenation for a total of 3 hr post-ROSC.”

Figure legend

15. Please spell out all abbreviations within the Sources of Funding section.

Response: We added the full funding information as recommended

1. National Institute of Environmental Health Sciences R21ES031243 (Jang)

2. National Heart, Lung, and Blood Institute R01HL141386 (Kilbaugh)

3. National Heart, Lung, and Blood Institute R56HL158696 (Jang)

4. National Heart, Lung, and Blood Institute R01HL166592 (Jang, Baker)

5. Children’s Hospital of Philadelphia Frontier Program (Baker, Ko, Forti, Kilbaugh)

6. National Institute of Neurologial Disorders and Stroke R01NS113945 (Baker, Lewis, Forti, Kilbaugh)

7. U.S. Department of Defense DoD W81XWH-22-1-0887/8 (Baker, Ko, Forti, Kilbaugh)

8. Toyota Way Forward Fund (Baker, Ko, Forti, Kilbaugh)

16. Please revise the figure to include the timing of each intervention.

Response: We revised the figure caption to include the timing of each intervention.

 

Reviewer #2: The proposed study is highly relevant with a very well described method. It will allow a very comprehensive evaluation of the effect of CO after cardiac arrest.

1. The submission of the design of experimental study should be encouraged. However, I do not know whether method paper could comply to Plos One Editorial policy, since the journal guidelines state that research articles "must present the results of original research". In order to include original findings, the article should at least present preliminary original data with a natural history of the model in control conditions.

Response: As editor mentioned, this submission is appropriate for the protocol section.

---

## [Decision Letter · Decision Letter 1]

8 Mar 2024

PONE-D-23-24722R1Carbon monoxide as a cellular protective agent in a swine model of cardiac arrest protocolPLOS ONE

Dear Dr. Jang,

Thank you for submitting your manuscript to PLOS ONE. After careful consideration, we feel that it has merit but does not fully meet PLOS ONE’s publication criteria as it currently stands. Therefore, we invite you to submit a revised version of the manuscript that addresses the points raised during the review process.

The Academic Editor is satisfied with scientific evaluation of your manuscript, and with the revisions you have made. Before we can can proceed with publication, we ask you to address the following concern: During internal evaluation of your manuscript, editorial staff noted significant text overlap between your manuscript and previous publications by one or more of the authors of this submission, specifically with the following publications: 1) "Alteration in Cerebral Metabolism in a Rodent Model of Acute Sub-lethal Cyanide Poisoning" - https://doi.org/10.1007/s13181-022-00928-w2) "Preliminary Research: Application of Non-Invasive Measure of Cytochrome c Oxidase Redox States and Mitochondrial Function in a Porcine Model of Carbon Monoxide Poisoning" - https://doi.org/10.1007/s13181-022-00892-53) "Cerebral mitochondrial dysfunction associated with deep hypothermic circulatory arrest in neonatal swine" - https://doi.org/10.1093/ejcts/ezx467 The Academic Editor has been consulted about this overlap, and they have not raised any concerns regarding duplication of work between these previous publications and this protocol submission, and they are satisfied that this protocol submission does warrant separate publication. However, while publication 3) above is published under an open access license, publications 1) and 2) are subject to copyright terms that prevent us from being able to publish your manuscript with the overlapping text in its current form. We apologize that this concern was not brought to your attention at an earlier stage in the publication process. In order to proceed, we ask you to revise your manuscript to reduce the text overlap with publications 1) and 2) linked above. Please also ensure, if you have not done so already, that the the previous publications are clearly referenced in the manuscript text as the sources for the methodologies included in this protocol. This is necessary for publication 3) as well. We appreciate your attention to these requests.

We look forward to receiving your revised manuscript.

Kind regards,

Hugh Cowley

Senior Editor

PLOS ONE

on behalf of

Meijing Wang, MD

Academic Editor

PLOS ONE

Reviewers' comments:

Reviewer's Responses to Questions

**Comments to the Author**

1. Does the manuscript provide a valid rationale for the proposed study, with clearly identified and justified research questions?

Reviewer #2: Yes

2. Is the protocol technically sound and planned in a manner that will lead to a meaningful outcome and allow testing the stated hypotheses?

Reviewer #2: Yes

3. Is the methodology feasible and described in sufficient detail to allow the work to be replicable?

Reviewer #2: Yes

4. Have the authors described where all data underlying the findings will be made available when the study is complete?

Reviewer #2: Yes

5. Is the manuscript presented in an intelligible fashion and written in standard English?

Reviewer #2: Yes

6. Review Comments to the Author

You may also provide optional suggestions and comments to authors that they might find helpful in planning their study.

Reviewer #2: This is perfect if it can go to a protocol section.

7. PLOS authors have the option to publish the peer review history of their article (what does this mean?). If published, this will include your full peer review and any attached files.

Reviewer #2: No

---

## [Author Response · Author response to Decision Letter 1]

25 Mar 2024

Thank-you for the comments related to overlap for the two papers list below (There was not concern for the third paper list above and we were not asked to address any overlap related to the third paper):

1) Reference 47-"Alteration in Cerebral Metabolism in a Rodent Model of Acute Sub-lethal Cyanide Poisoning" -https://doi.org/10.1007/s13181-022-00928-w

2) Reference 29- "Preliminary Research: Application of Non-Invasive Measure of Cytochrome c Oxidase Redox States and Mitochondrial Function in a Porcine Model of Carbon Monoxide Poisoning" - https://doi.org/10.1007/s13181-022-00892-5

Response: We reviewed our manuscript as well as the two papers above (published by our group). For ease for the editor, we make comments regarding changes in the manuscript related to overlap of the two list papers.

As also recommended by the reviewer, the list papers are already cited in the reference as seen above.

Introduction/background 

1. There is not overlap with the introduction/background as this paper focused on cardiac arrest and the application of carbon monoxide as a therapeutic which is unique and not discussed in the above papers. We did make small grammatical revisions to improve the paper but no major edits were made. 

Methods

1. This paper describes many common methods also found in the two listed paper. For example, we describe methods to measure respirometry in this manuscript which was also employed in the two list papers. While it is not considered plagiarism to describe the same methods, we did revise in this current manuscript so it is more brief and also reference the original papers. We specifically revised the following sections due to the overlap:

 a. Non-invasive optical monitoring of cerebral oxygenation and CIV redox rates

 b. Measurement of mitochondrial respiration in isolated brain mitochondria

 c. Western Blot 

 d. ATP fluometry

Discussion

1. There is not overlap with the discussion as this paper focused on cardiac arrest and the application of carbon monoxide as a therapeutic which is unique and not discussed in the above papers. We made edits to improve overall clarity of the manuscript and streamline the reliability.

---

## [Editor Report · Decision Letter 2]

4 Apr 2024

PONE-D-23-24722R2Carbon monoxide as a cellular protective agent in a swine model of cardiac arrest protocolPLOS ONE

Dear Dr. Jang,

Thank you for submitting your manuscript to PLOS ONE. After careful consideration, we feel that it has merit but does not fully meet PLOS ONE’s publication criteria as it currently stands. Therefore, we invite you to submit a revised version of the manuscript that addresses the points raised during the review process. Please submit your revised manuscript by May 19 2024 11:59PM. If you will need more time than this to complete your revisions, please reply to this message or contact the journal office at plosone@plos.org. Please include the following items when submitting your revised manuscript:A rebuttal letter that responds to each point raised by the academic editor and reviewer(s). You should upload this letter as a separate file labeled 'Response to Reviewers'.A marked-up copy of your manuscript that highlights changes made to the original version. You should upload this as a separate file labeled 'Revised Manuscript with Track Changes'.An unmarked version of your revised paper without tracked changes. You should upload this as a separate file labeled 'Manuscript'.If applicable, we recommend that you deposit your laboratory protocols in protocols.io to enhance the reproducibility of your results. Protocols.io assigns your protocol its own identifier (DOI) so that it can be cited independently in the future. For instructions see: https://journals.plos.org/plosone/s/submission-guidelines#loc-laboratory-protocols. Additionally, PLOS ONE offers an option for publishing peer-reviewed Lab Protocol articles, which describe protocols hosted on protocols.io. Read more information on sharing protocols at https://plos.org/protocols?utm_medium=editorial-email&utm_source=authorletters&utm_campaign=protocols.

We look forward to receiving your revised manuscript.

Kind regards,

Meijing Wang, MD

Academic Editor

PLOS ONE

Journal Requirements:

**Additional Editor Comments:**

The substantial revision to reduce the overlap with previous publications from the same group in Methods is appreciated. However, there is still minor concern. Some information is needed to help the readers better understand certain specific methods: 1) please provide one or two sentences to describe how CIV redox rates are determined; 2) how much tissue protein is used for Western Blot and for ATP fluorometry; and 3) The wavelengths for ATP fluorometry should be stated.

---

## [Author Response · Author response to Decision Letter 2]

6 Apr 2024

Thank-you for the comments. In our previous revisions we addressed the issue with overlap for the related to two of our publications listed below which satisfied the reviewer. 

1) Reference 47-"Alteration in Cerebral Metabolism in a Rodent Model of Acute Sub-lethal Cyanide Poisoning" -https://doi.org/10.1007/s13181-022-00928-w

2) Reference 29- "Preliminary Research: Application of Non-Invasive Measure of Cytochrome c Oxidase Redox States and Mitochondrial Function in a Porcine Model of Carbon Monoxide Poisoning" - https://doi.org/10.1007/s13181-022-00892-5

We were asked to address additional editor comments listed below. 

“The substantial revision to reduce the overlap with previous publications from the same group in Methods is appreciated. However, there is still minor concern.”

Some information is needed to help the readers better understand certain specific methods: 

1) Please provide one or two sentences to describe how CIV redox rates are determined

We added the following to the optics section:

Given that each tissue chromophore has a unique wavelength-dependent absorption signature, it is feasible to isolate the CIV influence on the brain's absorption spectrum from other tissue chromophores. We specifically used the UCLn algorithm to compute changes in cerebral oxidized CIV concentration from the measured wavelength-dependent changes in optical attenuation signals between 780–1000 nm wavelengths.

2) How much tissue protein is used for Western Blot and for ATP fluorometry 

For western blot:

Based on previous work we will use a protein concentration of 10 μl per well. All experiments will be performed in duplicates and the local background corrected density values.

For ATP fluorometry:

ATP concentrations will be obtained in snap frozen brain tissue samples (ratio of 10 mg of tissue to 100 µl of the assay buffer) using an ATP fluorometric assay kit (Sigma MAK190) with an excitation of 535 nm and emission of 587 nm obtained in a similar manner based on our previous publications [40, 47].

3) The wavelengths for ATP fluorometry should be stated.

We revised the ATP fluorometry to include the wavelengths based on our previous publications. The below is the revised paragraph.

ATP concentrations will be obtained in snap frozen brain tissue samples (ratio of 10 mg of tissue to 100 µl of the assay buffer) using an ATP fluorometric assay kit (Sigma MAK190) with an excitation of 535 nm and emission of 587 nm obtained in a similar manner based on our previous publications [40, 47].

---

## [Editor Report · Decision Letter 3]

10 Apr 2024

Carbon monoxide as a cellular protective agent in a swine model of cardiac arrest protocol

PONE-D-23-24722R3

Dear Dr. Jang,

We’re pleased to inform you that your manuscript has been judged scientifically suitable for publication and will be formally accepted for publication once it meets all outstanding technical requirements.

Kind regards,

Meijing Wang, MD

Academic Editor

PLOS ONE
---

## [Editor Report · Acceptance letter]

26 Apr 2024

PONE-D-23-24722R3 

PLOS ONE

Dear Dr. Jang, 

I'm pleased to inform you that your manuscript has been deemed suitable for publication in PLOS ONE. Congratulations! Your manuscript is now being handed over to our production team.

Kind regards, 

on behalf of

Dr. Meijing Wang 

Academic Editor

PLOS ONE